# Evaluation of Knockdown Bioassay Methods to Assess Phosphine Resistance in the Red Flour Beetle, *Tribolium castaneum* (Herbst) (Coleoptera: Tenebrionidae)

**DOI:** 10.3390/insects10050140

**Published:** 2019-05-16

**Authors:** Aaron Cato, Edwin Afful, Manoj K. Nayak, Thomas W. Phillips

**Affiliations:** 1Department of Entomology, Kansas State University, Manhattan, KS 66503, USA; aaronj.cato@gmail.com (A.C.); eddafful@umd.edu (E.A.); 2Department of Entomology, University of Arkansas, Fayetteville, AR 72701, USA; 3Department of Entomology, University of Maryland, College Park, MD 20742, USA; 4Department of Agriculture and Fisheries, GPO Box 46, Brisbane, QLD 4001, Australia; Manoj.Nayak@daf.qld.gov.au

**Keywords:** fumigation, hydrogen phosphide, narcosis, resistance management, stored products

## Abstract

Resistance to the fumigant phosphine in *Tribolium castaneum* occurs worldwide. This study evaluated tests based on adult knockdown time, the time for a beetle to become immobile, when exposed to a high concentration of phosphine. We recorded knockdown times of beetles that remained completely still for 30 s when exposed to 3000 ppm of phosphine in a large, gas-tight glass tube. Beetles were used from 12 populations, of which six were ‘susceptible’ to phosphine, three were ‘weakly resistant’, and three were ‘strongly resistant’. Knockdown times were determined for single beetles, as well as for groups of ten beetles for which the time to knockdown for either five beetles (KT_50_) or ten beetles (KT_100_) were recorded. Similar knockdown times occurred across susceptible and resistant populations. However, the KT_100_ tests generated conservative times for diagnosing strong vs. weak resistance. The strong resistant populations were all over 100 min with KT_100_, compared to 60 min or less for susceptible and weak resistant populations. Special tests on single beetles revealed higher knockdown times in insects that were deliberately disturbed compared to those without any disturbances. Work reported here suggests a knockdown test conducted on beetles in a matter of minutes or hours could help classify phosphine resistance status prior to decisions on phosphine fumigation.

## 1. Introduction

Insect pests infest grain that is stored for a few months to up to one year or longer, and fumigation is commonly performed for disinfestation [1]. One of the most cosmopolitan pests to cereal grains and milled grain products is the red flour beetle, *Tribolium castaneum* (Coleoptera: Tenebrionidae) [2]. This pest can be controlled through fumigation with hydrogen phosphide gas, also known as phosphine. However, populations of *T. castaneum* across the globe have developed resistance to phosphine. The mode of action for phosphine toxicity is not well-known, but likely includes disruption of the nervous system, suppressed energy metabolism and limitations on oxygen metabolism, all of which could have mutations that facilitate resistance [3]. Phosphine resistance for *T. castaneum* was first reported in the United States in 1976 [4], and confirmed by several studies in the past 30 years [5,6,7,8]. The widespread occurrence of phosphine resistance in *T. castaneum* and other pest species has motivated research to develop tests for resistance that can determine the presence or absence of resistance and thus help with pest management decisions [9].

Phosphine resistance can be reliably determined using a discriminating dose bioassay under controlled conditions. A working group of the United Nations Food and Agriculture Organization (FAO) developed FAO method number 16 [10] for diagnosing phosphine resistance in major stored product pests. The so-called FAO test uses a discriminating dose bioassay for adult insects of seven different stored product species, including *T. castaneum*. This bioassay for *T. castaneum* requires a 20 h fumigation with 30 ppm of gas at 25 °C, followed by a 14-day post-fumigation period in clean air to allow for recovery or delayed mortality, if any occur, in the tested sample [10]. Phosphine susceptible adult *T. castaneum* should all die at the discriminating dose, while any survivors would be considered resistant to phosphine. The discriminating dose is the estimated concentration needed to kill 99.9% of adult beetles from a susceptible population. While the FAO bioassay is reliable to detect resistance in a given population, it requires technical methods and analytical instruments to apply and analyze phosphine that are available only in specialized laboratories and not typically available to commercial grain storage facilities or pest control companies. In addition to the technical requirements, the FAO assay is impractical for users who need same-day information about pests to help make decisions about fumigation. These impracticalities have led to developing a simple one-day test for phosphine resistance, referred to by many as a ‘quick test’, that is practical, accurate and inexpensive for diagnosing phosphine resistance compared to the FAO assay [9].

One-day tests, so called “quick tests”, for phosphine resistance are not based on mortality but rather on a behavioral phenomenon referred to as “knockdown” or “narcosis”. A ‘knocked down’ beetle is one that stops walking, falls over on its back and does not return to an up-right position and/or ceases all observable movement of any body part when exposed to unusually high concentration of phosphine. When insects are exposed to phosphine at unusually high concentrations, such as 5 mg/L (>3000 ppm), they will enter narcosis within a given time relative to their resistance status. Susceptible insects will be knocked down within a few minutes of exposure while resistant insects would be knocked down after a longer exposure time [11,12,13,14]. Narcosis by definition is not mortality, as treated insects become active again after ventilation with fresh air [11,15,16].

Phosphine resistance in several species of stored grain pests can occur in two genetically controlled phenotypes, referred to as weak resistance or strong resistance [3,17,18]. The genetic bases for these phenotypes rely mostly on an insect being homozygous for specific genotypes at two different loci [3]. The toxicity metric called the “resistance ratio” value, the RR_50_, can help to classify resistant insects as having either the strong or the weak form of resistance. The RR_50_ is computed from dose-mortality data subjected to Probit analysis on a given population for which the LC_50_ in the resistant population is divided by the LC_50_ for a standard lab-susceptible population. RR_50_ values between about 10 and 50 may characterize a weak-resistant population, while RR_50_ values greater than 50 can characterize populations with strong resistance [9]. Work by Nayak et al. [19] reported a knockdown test for *Cryptolestes ferrugineus* (Coleoptera: Laemophloeidae) that distinguished between beetles that had either weak resistance or strong resistance to phosphine, and similar methods were recently developed to distinguish strong from weak resistant *Sitophilus oryzae* (Coleoptera: Curculionidae) [20]. Strong resistance in populations of *T. castaneum* was identified in our recent geographic survey in North America, and these had RR_50_ values up to 127 [8]. A quick test for diagnosing both weak and strong phosphine resistance in *T. castaneum* would be a useful tool for resistance management of this pest.

The work by Steuerwald et al. [21] led to the development of a test kit by Detia Degesch (Laudenbach, Germany) using knockdown times as diagnostics in evaluating stored grain insects for tolerance to phosphine. The Detia-Degesch kit (Figure 1) provides the user with small pellets of magnesium phosphide that can be added to water and generate about 6000 ppm phosphine gas in a collapsible plastic chamber. A large plastic transparent syringe is provided to observe behavior of beetles treated with a 3000 ppm dilution and record time to knockdown for comparison with diagnostic times given in the test kit’s literature. Recent work by Athanassiou et al. [22] evaluated the utility of the Detia-Degesch kit for knockdown responses of *T. castaneum* at two phosphine concentrations and a sequence of hold times. That study provides key knockdown results with phosphine for two laboratory strains, one susceptible and one resistant to phosphine, and complements our work reported here.

The overall goal of our research was to assess and improve the ability of a knockdown test for phosphine resistance in *T. castaneum* as reliably as the FAO test with a diagnostic lethal dose. Our first objective was to compare the results of recent FAO discriminating dose bioassays conducted on several populations with knockdown tests on single beetles or groups of beetles from the same populations. Our second objective was to assess the importance of physical disturbance of tested beetles on beetle knockdown time.

## 2. Materials and Methods

### 2.1. Tribolium castaneum Populations

For all quick test bioassays performed, insects were taken from laboratory cultures of selected *T. castaneum* populations studied in the research reported by Cato et al. [8] and Chen et al. [18] and listed in Table 1, with one exception. The population named “Thailand” (Table 1), which was from a long-term laboratory colony at the USDA ARS laboratory, was originally collected in Thailand over 40 year ago. The phosphine resistance status of the Thailand strain was unknown so we subjected beetles to the FAO discriminating dose test using the same methods as in Cato et al. [8], and estimated a resistance frequency of 15%. Selection of colonies chosen for quick test assessment was based on frequency of resistance determined with the FAO test conducted previously by Cato et al. [8]. The phenotypes, i.e., susceptible, weak, or strong resistance, were determined by Cato et al. [8] and we made provisions for adequate numbers of beetles in cultures for the current experiments. The knockdown experiments reported below used cultures of the populations described here that were within 5 generations after their use in the Cato et al. [8] experiments, or, in the cases of the Russell and Mitchell populations, up to two years (20 generations) when last studied by Chen et al. [18]. Beetles were reared in 1.0 L glass canning jars with a screw-top lid ring and a metal screen combined with filter paper was in place of the supplied metal lid insert. Beetles were reared on organic golden buffalo flour supplemented with 5% brewer’s yeast (95:5). Colonies were kept in growth chambers set at 28 °C and 65% relative humidity with a photoperiod of 16 h light and 8 h dark.

### 2.2. Knockdown Definition 

We conducted preliminary trials to determine which type of knockdown score would generate the least variation among people conducting the assay and that can be described with clear written instructions. We chose not to use descriptive terms referring to treated insects as being simply “hampered”, “inactive”, or “unable” to walk [12,15,21]. We define “knockdown” for our purposes as the persistent and complete lack of movement by any body part of a treated insect. We observed in some cases, especially those with multiple beetles in a test chamber, that a knocked down beetle might resume movement in a matter of seconds. We therefore modified our criterion that a beetle must be motionless for 30 sec before it was scored as knocked down. The knockdown time for a group of beetles was recorded only after the last beetle in the group was motionless for 30 sec. This 30-sec knockdown criterion was used when scoring the time to knockdown of a single beetle, the first 5 beetles in a group of 10, or all 10 beetles in a group (see below).

### 2.3. Methods to Conduct and Assess Knockdown Tests

We chose the general methods used for the Detia-Degesch test kit as elaborated by Steuerwald et al. [21]. We conducted experiments with a variation of the Detia-Degesch assay so that our knockdown assessment methods could be used to improve efficiency. The earlier work determined that 3000 ppm of phosphine was an acceptable level of phosphine that was not too high or too low for generating a useful level of narcosis, and it could easily be reached using the magnesium phosphide pellets found within the commercial kit [21].

Knockdown assays were conducted on a laboratory bench in a room maintained at 25 °C with standard white lighting. Rather than the large plastic syringe for a knockdown chamber, a PYREX^®^ 55 mL Screw Cap Culture Tubes with PTFE-lined phenolic caps, 25 × 150 mm, were used as fumigation chambers (Figure 2). Fisherbrand™ Turnover (Fisher Scientific, Loughborough, UK) septum stoppers were used at the opened end of the tubes to make them gas tight and allow for introduction of phosphine (Figure 2). Beetles were placed inside these large tubes with clean air and no food or other substrate, and the tube was sealed with the large septum. A volume of 16.5 mL of 1% phosphine was added to the fumigation tubes with a gas-tight syringe (Hamilton^®^ Fisher Scientific; 25 mL, Model 1025 TLL) after first removing an equal volume of air. This dilution of 1% phosphine was determined to bring the fumigation chambers to approximately 3000 ppm, the same concentration used in the Detia-Degesch test kit [21]. Once the gas mixture was added to the fumigation tubes we analyzed the phosphine concentration using quantitative gas chromatography with a flame photometric detector as we described previously [8]. Any tube with a phosphine concentration that varied more than 1% was removed from the study.

Three methods of knockdown tests were evaluated with the 12 *T. castaneum* populations listed in Table 1. We evaluated each method for its utility in diagnosing phosphine resistance among the populations, and to determine which of the three methods gave the fastest answer about resistance for each population. We compared the time to knockdown of single individuals from a given population, the time to knockdown of 50% (KT_50_) of a sample of ten insects from a population, and the time to knockdown of 100% (KT_100_) of a sample of ten insects from a population. Both the time to 50% and 100% knockdown of a population are the traditional methods used in knockdown tests. Single insect knockdown was included mainly for comparison, as it was decided to test efficiency of the test with an undisturbed single beetle, rather than assume any group or sub-group of multiple beetles gave similar or more accurate estimates of results. Single insect trials were tested by using 1 insect per replication (*n* = 10) for the 12 different populations. Time to knockdown was determined using methods as described above, which was the time that a beetle had a complete lack of movement for at least thirty seconds. For both KT_50_ and KT_100_ trials, 10 insects were used per replication (*n* = 5) with a complete lack of movement for at least thirty seconds for either 50% or 100% of the beetles scored as knocked down. The USDA lab strain is the experimental control as it is a known susceptible strain against which all other strains can be compared. 

### 2.4. Evaluation of Physical Disturbance on Beetle Knockdown Time

To better understand the basis for shorter knockdown times we observed in single insect trials compared to knockdown times recorded with groups of beetles from the same population, we studied the role of an artificial stimulus applied to a single beetle during a knockdown trial. We used insects from two separate populations for this study, the USDA strain that was classified as susceptible with the FAO test, and the Red Level population to represent a strong resistant population [8], to study the effect of disturbance. Bioassay tubes with a single beetle and containing 3000 ppm of phosphine were subjected to two different treatments. Either the vial was not moved at all during the full duration of the knockdown test, or at every minute the vial was rolled 360° in one direction and then 360° back to its starting place in a period of about 5 sec. We decided on this standard rolling method after preliminary studies on beetle disturbance showed that simple movement of vials on the lab bench would increase knockdown times. Pairs of stimulus and non-stimulus trials were performed to determine if this phenomenon was consistent in both susceptible (*n* = 40) and resistant populations (*n* = 20), and if the phenomenon would create a statistically significant difference in knockdown times.

### 2.5. Statistics

The R statistical program was used for all statistical analyses [23]. For the comparison of knockdown times in the single beetle trials, the KT_50_, and the KT_100_ we used a two-way ANOVA with a post-hoc Tukey’s HSD test to identify significant variation in raw sample data between levels of the treatments within a knockdown test type, and the interaction of different treatments and populations. Time to knockdown was also compared across the three test methods for each population, which were previously determined to have a level of resistance categorized by the FAO test for each populations as determined by Cato et al. [8]. We analyzed experiments comparing knockdown times for single beetles that were either disturbed or not disturbed using unpaired T-tests. Here we studied two populations that represented fully susceptible or strong resistance status, and their results were analyzed separately.

## 3. Results

### 3.1. Variation in Knockdown Times among Populations with Different Resistance Levels

The tests using single beetles had knockdown times ranging from 8.46 to 52.70 min with considerable overlap among populations previously determined to be either resistant or susceptible in prior FAO testing (Table 1). There was also considerable overlap among resistant and susceptible populations for the knockdown times in the experiments using the 50% and 100% criteria. The beetles from the Red Level population, previously determined to have 100% resistance using the FAO test, were the only samples to have significantly longer knockdown times in each type of knockdown assay when compared with the five susceptible populations, those with FAO values at 0% resistance. Analyses of variance comparing the three quick test treatments within each population showed significant differences for all comparisons (comparing across columns in Table 1). However, means comparison for all populations but the USDA strain showed no differences between knockdown times recorded with the single beetle tests and the 50% tests, but that the knockdown times for the 100% tests were significantly longer than the other two.

### 3.2. Effect of Physical Disturbance on Knockdown Times of Single Beetles

We observed a significant effect of disturbance on knockdown times of single beetles (Figure 3). For the susceptible population from USDA, the mean knockdown time for beetles without any stimulus applied to the assay vial was 7.43 min (SE = 0.31; *n* = 40) minutes, while beetles with a stimulus added took 8.81 min (SE = 0.31) minutes to knockdown. For the resistant population from Red Level, the mean knockdown time for beetles without stimulus was 13.03 min (SE = 1.16) (*n* = 20) minutes, and 32.94 min (SE = 3.47; *n* = 40) minutes for trials with an added stimulus (Figure 3).

## 4. Discussion

Our results provide a baseline for designing a useful same-day test for phosphine resistance in *T. castaneum* based on the needs of the user. We found that the single knockdown test and KT_50_ were not statistically different in all but one of the populations we tested and that the knockdown times for KT_100_ tests were statistically separated from the first two tests for each of those populations. Knockdown times for the KT_100_ tests were consistently three-fold or longer than times for the other two tests. Such information on consistent trends within and among the knockdown times for these tests could influence selection of any one of these for regular use in fumigation decisions based simply on the time to conduct a given test. However, the criterion for selecting any simple test for phosphine resistance should be for a test’s reliability. We used six resistant population of *T. castaneum* for our experiments. Three of these populations, Thailand, Abilene, and Russell, had the common “weak resistant” status, with RR_50_ values between 10-fold and 50-fold, while the other three populations, Minneapolis, Mitchell, and Red Level, were characteristic of strong-resistance populations, having RR_50_ values over 100-fold based on Probit analyses [8]. Each of the three knockdown tests had considerable overlap in knockdown times across susceptible and weak-resistance populations, with no significant differences separating them except for the one case of KT_50_ for the Russell population being significantly longer than the longest knockdown time for any of the susceptible strains. Additionally, we found considerable overlap in knockdown times for the three tests in comparing weak-resistance from strong-resistance populations. The ability to separate resistant from susceptible populations is critical for decisions on fumigations to control grain pests, and our results with knockdown tests for *T. castaneum* suggest that more work is needed to develop a reliable quick test.

Despite our results showing that knockdown tests can be unreliable to some degree in diagnosing *T. castaneum* as phosphine resistant, a conservative decision on a suitable knockdown time for susceptibility could be proposed from our data. The three tests show overlap with no statistical difference across the six susceptible populations. However, the KT_50_ times for the first three susceptible populations in Table 1 could support a KT_50_ of 7 min or less to assign a phosphine susceptible status. KT_50_ times overlap considerably for the five populations of Davis, Arbuckle, Williams, Thailand and Abilene. The KT_50_ time for Abilene at 11.50 min is statistically longer than the 7.27 min for Calgary in our dataset, which suggests 7 min may be an upper limit below which a population could be classified as susceptible. However, Thailand’s KT_50_ at 7.85 min being similar to that of Calgary weakens the reliability of a 7 min threshold for a population classified as susceptible. Our data could support using a KT_50_ time of 6 min as reliable for a susceptible designation in a large majority of cases. Future research with more samples of beetles from more populations of *T. castaneum* could add information on the reliability of the 6 min KT_50_ threshold to separate phosphine susceptible from resistance samples.

Knockdown tests to distinguish strong-resistance *T. castaneum* populations from those having weak resistance have similar challenges as those for a diagnostic tool to separate susceptible from resistant populations, though conservative compromises are possible. Table 1 shows considerable overlap across weak and strong-resistant populations in all three tests. The Minneapolis and Mitchell populations, both previously described as being strongly resistant using a FAO-type diagnostic test [18], were not consistently different from one or more weak-resistance populations in the single-beetle and KT_50_ knockdown tests. However, the Red Level population, scored strongly resistant by Cato et al. [8] based on a RR_50_ value of 127.11, had knockdown times recorded in all three tests that were present only in the group of three strongly resistant populations with no overlap with populations in the weakly resistant group. Therefore, based on our data, a conservative lower threshold for strong resistance could be 60 min for the single-beetle KT test, 30 min for the KT_50_ test, and perhaps 180 min for the KT_100_ test. Our results support those by Athanassiou et al. [22] showing that 3000 ppm applied to a strongly resistant strain of *T. castaneum* from Brazil resulted in more than half the beetles in a group of 20 remaining active after 90 min of exposure.

Our experiments reported in Figure 3 show that the knockdown time for a single beetle exposed to a narcotic concentration of phosphine can be increased by treatment of the single beetle with a physical disturbance. The disturbance we provided with the rolling of the treatment tube was to simulate beetles in a group being moved around by repeatedly contacting each other, or falling over as they touch each other when confined to a small space. The differences in knockdown times between stimulated and unstimulated beetles were statistically significant for both populations tested. However, the difference between stimulated and unstimulated single beetles from the phosphine susceptible population was much smaller than the difference found with the resistant population. In the resistant population, the time-to-knockdown was nearly three times higher when there was a brief periodic stimulus compared to the knockdown times for unstimulated beetles. The stimulus-driven knockdown time of the resistant population in single trials was similar to that of other populations with the same resistance level for the KT_100_ (Table 1). We speculate that the increased time to complete narcosis from a high concentration of phosphine is similar to that in many animals approaching a period of sleep that are being disturbed in the process. Sleep or toxin-induced narcosis will happen faster when the animal is not disturbed. The long knockdown time for the resistant beetles, whether undisturbed or stimulated, likely represents the nature of phosphine-resistant insects being less intoxicated by phosphine compared to susceptible insects. The role of disturbance in knockdown bioassays for phosphine resistance should be considered in cases whereby beetles are allowed to continually crawl up a surface and fall compared to assays in which insects are only allowed to move horizontally with little disturbance [15,24].

The potential for weak and strong resistance to phosphine in many populations of *T. castaneum* requires that pest managers have a relatively non-technical quick test to diagnose resistance status of beetle populations needing mitigation. Results from our study suggest a number of recommendations for further development and use of Steuerwald et al.’s [21] commercial quick test, or similar knockdown kits, for such diagnoses. A clearly stated and easily understood definition of knockdown, as what we used here with the 30 sec hold time, should decrease ambiguity in any test for narcosis by different people. The paper by Steuerwald et al. [21] simply says “Every minute note the number of the knocked down beetles…” to describe the insects behavior, “knocked down”, that the observer should record, but the instructions give no more details for evaluating knockdown. Our single-beetle knockdown tests were no better in assessing resistance in *T. castaneum* than were the groups of beetles, but a confident decision based on single beetles should require several single-beetle assays be conducted. We suggest that two or three group assays of 10 beetles can generate good information that represents a test population better than would a lengthy series of single beetle tests. Either a KT_100_ or KT_50_ with reliable diagnostics for susceptible and resistant insects should be determined in future work to provide a useful tool for grain managers. It is clear from toxicity assays and in some limited field observations [7,8,17] that pest populations with low frequency levels of weak resistance can be controlled with phosphine in the field with proper high gas concentrations and long hold times. It is the strongly resistant pest populations that pose the biggest challenges, and these populations need to be identified early on so that a decision can be made for phosphine alternatives rather than continuing with phosphine that will likely fail [9]. Concerns for controlling strongly resistant grain pests may then call for a same-day knockdown test kit to confidently warn of a strong phosphine-resistant pest population, with the knowledge that samples with shorter knockdown times may represent either susceptible or weak-resistance insects.

## 5. Conclusions

This work shows that *T. castaneum* adults can be evaluated over a period of minutes to hours for their phosphine-resistance status by determining the time to knockdown in a bioassay with 3000 ppm of phosphine in an observation chamber. However, we studied lab strains of populations with a known resistance status previously determined with a reliable two-week diagnostic dose assay, and we found no clear division in knockdown times between susceptible and resistant populations. Average knockdown times using either a single beetle or a group of beetles varied broadly among several susceptible and resistant populations. However, the average knockdown times for beetles in a group of 10 were markedly longer for strong-resistant beetles, from 100 to 194 min, while knockdown times for susceptible and weak-resistance populations varied from 17 to 63 min. Longer times to knockdown for beetles tested in a group were attributed to the disturbance experienced by beetles touching each other as they walked in the chamber. Controlled experiments with our lab-susceptible strain and our strongest-resistance population used a single beetle in a chamber that was either physically disturbed or not disturbed. We found longer knockdown times for beetles that were disturbed by rolling the chamber compared to beetles in an undisturbed chamber. The disturbance of groups of moving beetles in a test may therefore result in a so-called quick test that is longer than 30 min, but could be useful for identifying populations with strong resistance based on times longer than 100 min. Strongly phosphine-resistant beetles infesting grain storages can be difficult to control with well-done fumigation. A reliable same-day quick test could be very useful for fumigators and grain managers to assess the resistance status in a field population of *T. castaneum*, and the results reported here suggest there is potential for developing such a method with a simple knockdown test.

## Figures and Tables

**Figure 1 insects-10-00140-f001:**
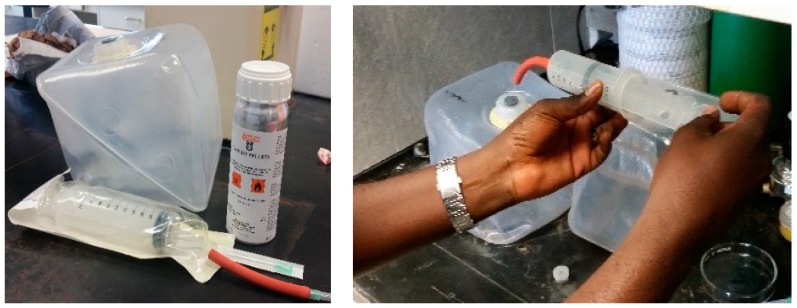
The Detia-Degesch knockdown test kit. The components of the kit (left) are: A collapsible 3.8 L plastic container in which the magnesium phosphide tablet is combined with water to generate about 6000 ppm of phosphine upon reaction; the unopened plastic syringe for use in exposing beetles to gas; and the gas-tight metal container storing the individual sealed tablets of magnesium phosphide. On the right is the transparent plastic syringe with test beetles inside while being loaded with the 6000 ppm phosphine prior to drawing in an equal volume of fresh air to dilute the chamber concentration to about 3000 ppm.

**Figure 2 insects-10-00140-f002:**
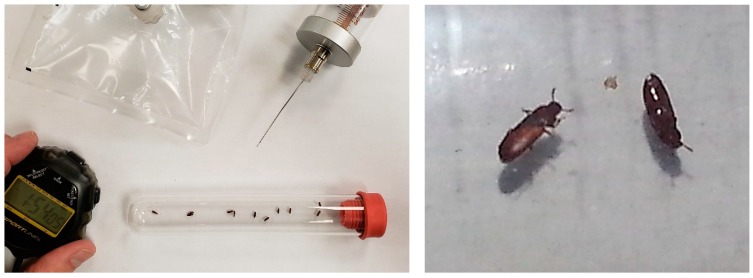
The photo on the left shows the glass exposure tube into which beetles were placed for the knockdown assays. The tube was closed with a large red injection septum, air was removed, and then a high concentration of phosphine from the holding bag was injected with the large glass gas-tight syringe to achieve 3000 ppm. The stopwatch was started to measure time to knockdown after the phosphine was added. On the right are two *T. castaneum* undergoing knockdown narcosis inside the glass tube with phosphine. The beetle to the left is upright and walking unsteadily, while the beetle to the right has fallen on its back and is moving its appendages.

**Figure 3 insects-10-00140-f003:**
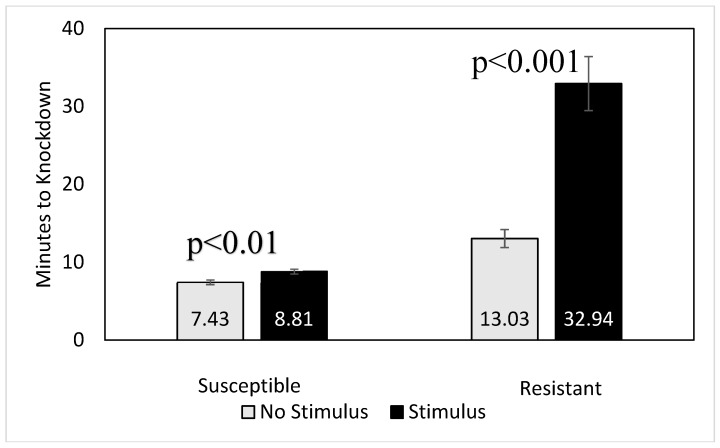
Mean (SE) time to knockdown of individual *Tribolium castaneum* from susceptible and resistant populations exposed to approximately 3000 ppm of phosphine while stimulated or not stimulated during the observation time. *p*-values are those for *t*-test analyses for comparison of knockdown times for physically disturbed and undisturbed beetles.

**Table 1 insects-10-00140-t001:** Mean knockdown time (KT) in minutes for populations of adult *Tribolium castaneum* assayed as single beetles or as groups of 10 for knockdown of 50% or 100% of the group.

Population ^1^ & Resistance Status	FAO % Resist. ^2^	Single KT (± SE) ^3,4,5^			50% KT (± SE) ^3,4,5^			100% KT (± SE) ^3,4,5^		
Susceptible										
USDA Lab Susceptible	0	8.46	±	0.66 A,a	5.19	±	0.21 A,b	9.13	±	0.46 A,a
Excelsior Springs	0	10.82	±	1.04 A,a	6.13	±	0.48 A,a	17.05	±	3.01 AB,b
Calgary, Alberta	0	9.24	±	0.42 A,a	7.27	±	0.11 A,a	16.23	±	1.21 AB,b
Davis, Calif.	0	14.13	±	0.84 AB,a	9.16	±	0.84 AB,a	34.83	±	7.42 AB,b
Arbuckle, Calif.	0	14.37	±	1.35 AB,a	10.00	±	0.30 B,a	31.28	±	3.12 AB,b
Williams, Calif.	0	10.91	±	1.01 A,a	10.05	±	0.53 B,a	26.16	±	2.15 AB,b
Weak Resistance										
Thailand	15	14.62	±	2.00 AB,a	7.89	±	0.29 AB,a	29.17	±	3.92 AB,b
Abilene, Kansas	39	16.14	±	1.21 AB,a	11.50	±	0.61 BC,a	37.30	±	3.84 AB,b
Russell, Kansas	41	16.82	±	1.38 AB,a	14.38	±	0.42 CD,a	62.78	±	9.12 BC,b
Strong Resistance										
Minneapolis, Kansas	89	24.47	±	2.35 B,a	16.98	±	1.04 D,a	100.20	±	11.77 C,b
Mitchell, Kansas	93	19.32	±	1.54 AB,a	14.55	±	0.87 CD,a	172.15	±	24.72 D,b
Red Level, Alabama	100	52.70	±	1.54 AB,a	24.33	±	2.05 E,a	194.13	±	22.13 D,b

^1^ Name/location of each population tested and its resistance status as per Cato et al. [8]. ^2^ Percent of beetles in a population estimated to be phosphine-resistant determined in a recent FAO test [8]. ^3^ Replications per KT type as follows: Single (10), KT_50_ (5), and KT_100_ (5). ^4^ Means for KT results in a row followed by the same lower case letter, and those in a column followed by the same upper case letter, are not significantly different according to a Tukey HSD post hoc analysis, *p* < 0.05. ^5^ All differences were determined by a Two-Way ANOVA, F = 8.416, df = 22, 204, *p* < 0.01.

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
