# Peer review of "Evaluation of Knockdown Bioassay Methods to Assess Phosphine Resistance in the Red Flour Beetle, Tribolium castaneum (Herbst) (Coleoptera: Tenebrionidae)"

_insects, 2019, doi:10.3390/insects10050140_

Round 1
Reviewer 1 Report
Comments for Authors:
The manuscript describes a newer method for assessing phospine resistance in several different populations of T. castaneum. The authors describe in detail the use of traditional FAO methods and the down-sides of using those techniques. They then describe a new method that could be used to distinguish between susceptible and resistant populations. The methods are quite extensive but are necessary to lay out all of the appropriate comparisons. The results, on the other hand, are short but are simply the results of 4 comparisons: single beetle knock-down time; KT50 in groups, KT100 in groups; and single beetles with and without disturbance. Although I would have liked a graphical result better than the Table results, they are presented fine. I would suggest that the Table appear closer to the results section in the publication, as I was confused as to why there was no Table near the results. Discussion of the work is good too and presents the results with caution, but also presents some nice discussion of using these results as a good place to make some tentative thresholds for KT in populations.
Specific comments:
Line 30: I didn’t know what you meant by disturbance when I first read this in the abstract. Try to define what you mean before you present the results.
Lines 18, 42, 88, 143, 193, 304: A few grammatical errors in these sentences. Please read through carefully.
Figure 1 caption: The ordering of the photos is odd as described in the caption. Just make sure that when the final version comes out, the figure is as you want it to look and is as you describe in the caption.
Lines 142-145: It’s not totally clear to me what is different about the Thailand strain. Is it newly added to these analyses? I think that is the case, but I’m not sure.
Figure 2 caption: I would delete the final sentence of this caption, since the photo seems to show multiple beetles within the tube but the experiment you described with disturbance on single beetles, would only have 1 beetle within the tube.
Lines 228-230: Final sentence of this paragraph can be deleted. It is redundant.
Line 239: I think bases should be basis.
Lines 239-241: Contradicts what was said in the abstract and the results.
Results: Where are the GC results? If you don’t present them you should delete the information from the methods.
Line 305: Please use the same label for a population throughout the paper, particularly, Thai versus Thailand.
Discussion: What about presenting these results as more of a baseline for testing. You do this towards the end of the discussion but maybe this is best for resistant populations only and then if populations are on the border of the threshold, you use a more safisticated test, like the FAO.
Line 338: Missing a word in: “three strongly populations”
Line 362: Extra “phosphine” in this sentence.
Discussion: You mention in the introduction a reference to Athanassiou et al 2019 but do not talk about it much in the discussion. Can you expand on how your results compliment each other?
References: Check formatting of references. I think they should be numbered.
Author Response
Dear Reviewer 1. Thank you for your helpful comments. Please refer to my response to you, written in bold and indented on the attached Word file. I am also copying below, just in case the file is not attached....
*********************************
Reviewer 1
Comments and Suggestions for Authors
Comments for Authors:
The manuscript describes a newer method for assessing phospine resistance in several different populations of T. castaneum. The authors describe in detail the use of traditional FAO methods and the down-sides of using those techniques. They then describe a new method that could be used to distinguish between susceptible and resistant populations. The methods are quite extensive but are necessary to lay out all of the appropriate comparisons. The results, on the other hand, are short but are simply the results of 4 comparisons: single beetle knock-down time; KT50 in groups, KT100 in groups; and single beetles with and without disturbance. Although I would have liked a graphical result better than the Table results, they are presented fine. I would suggest that the Table appear closer to the results section in the publication, as I was confused as to why there was no Table near the results. Discussion of the work is good too and presents the results with caution, but also presents some nice discussion of using these results as a good place to make some tentative thresholds for KT in populations.
We now moved Table 1 to the Results section. We now rely on the journal editors to assure it is in the proper location when published.
Specific comments:
Line 30: I didn’t know what you meant by disturbance when I first read this in the abstract. Try to define what you mean before you present the results.
We revised this explanation in the Abstract to be clear for the reader; now lines 29-33.
Lines 18, 42, 88, 143, 193, 304: A few grammatical errors in these sentences. Please read through carefully.
We corrected sentences at these line numbers for grammar as needed.
Figure 1 caption: The ordering of the photos is odd as described in the caption. Just make sure that when the final version comes out, the figure is as you want it to look and is as you describe in the caption.
We revised Figure 1 to have just the first two photos. Both are appropriate for the Introduction because they show the commercial test-kit which was the model for our experimental system.
Lines 142-145: It’s not totally clear to me what is different about the Thailand strain. Is it newly added to these analyses? I think that is the case, but I’m not sure.
We revised this sentence to be clear that the resistance status of this strain was unknown, and we evaluated it prior to its use in the current study.
Figure 2 caption: I would delete the final sentence of this caption, since the photo seems to show multiple beetles within the tube but the experiment you described with disturbance on single beetles, would only have 1 beetle within the tube.
We revised Figure 2 with a new photo of the experimental set-up that clearly shows, and refers to a knockdown tube containing a group of beetles, the large syringe for introducing gas, the bag from which the gas was drawn and the hand of an observer with a stopwatch. Our tubes were always observed flat on the lab-bench, not at an angel as shown in the original photo. The second photo is the one showing a close-up of two partially knocked down beetles. The revised figure is now at about ll. 216-223.
Lines 228-230: Final sentence of this paragraph can be deleted. It is redundant.
Agreed; the sentence is deleted.
Line 239: I think bases should be basis.
It is changes to “basis”.
Lines 239-241: Contradicts what was said in the abstract and the results.
True. Re replaced “longer” with “shorter”, so that we are consistent in talking about single beetles being knockdown in a shorter time than beetles in groups.
Results: Where are the GC results? If you don’t present them you should delete the information from the methods.
GC analyses were done simple to verify that the knockdown tubes were properly filled with 3000 ppm of phosphine. We agree there is more narrative than needed here. Tubes that were improperly dosed were not used. We deleted that full paragraph titled “GC Analysis” and have inserted a brief description and literature citation for GC validation of gas at the end of the second paragraph in “Assessment of Quick Test Techniques”.
Line 305: Please use the same label for a population throughout the paper, particularly, Thai versus Thailand.
Corrected
Discussion: What about presenting these results as more of a baseline for testing. You do this towards the end of the discussion but maybe this is best for resistant populations only and then if populations are on the border of the threshold, you use a more safisticated test, like the FAO.
We now have a first sentence in the Discussion stating that our results provide a baseline for selection of a useful 1-day test based on needs of the user. As the reviewer states, we bring up the “baseline” idea in other parts of the Discussion, so it is important to mention it at the start of the Discussion.
Line 338: Missing a word in: “three strongly populations”
The word “resistant” was inserted after “strongly”.
Line 362: Extra “phosphine” in this sentence.
Yes. We deleted the first “phosphine”.
Discussion: You mention in the introduction a reference to Athanassiou et al 2019 but do not talk about it much in the discussion. Can you expand on how your results compliment each other?
We now have a second citation Athanassiou on about ll. 368-370 to point oit that our results support those of that paper.
References: Check formatting of references. I think they should be numbered.
Citations and References are now properly formatted and numbered. We trust that the copy-editors of the journal will confirm this, if and when the paper moves to publication.

Reviewer 2 Report
General comments:
1- This work has a serious importance in the field of stored products. The researchers tackled with an important problem related to identify phosphine resistant populations of T. castaneum.
2- In the tested methods there are no control groups of insects. I wander what happens if due to experimental errors insects are subjected to accidental toxic substances that may cause knockdown? How reliable will be the results? The authors are invited to explain why in those tests control groups can totatlly be ignored.
Specific comments:
Line 18: …"become immobilize"… shouldn't be …"become immobilize"…?
Line 75: …"as 5 mg/day (>3000 ppm),"… 3000 ppm is concentration has nothing to do with the exposure time. It should be "…5 mg/m3 (>3000 ppm),…"
Line 87: Schlipalius et al., 2012; is not on the list of references.
Line 89: Schlipalius et al., 2012; is not on the list of references
Line 216:…"… our 12 T. castaneum populations…" the 12 T. castaneum needs explanation.
Lines 251 and 252: The sentence "… An equal number of trials for added stimulus and no stimulus within each population were…." needs rephrasing.
Line 273: "…ranging from 8.46-52.70…" should be : "…ranging from 8.46 to 52.70…"
Lines 278 and 279: "…in each QT assay type…" what is QT essay type? needs explanation.
Lines 324 and 326: "…KT50 …" should be "…KT50 …"
Line 338: "…three strongly populations…" should be "…three strong populations…"
Lines 445-447 "Waterford, C. J., and R. G. Winks. 1994. .." is not in text.
Author Response
Dear Reviewer 2,
Please see the attached Word document with my responses to your comments and suggestions. I have also pasted the same file below, for security. Thank you for your helpful review. Tom Phillips
*****
Reviewer 2
General comments:
1- This work has a serious importance in the field of stored products. The researchers tackled with an important problem related to identify phosphine resistant populations of T. castaneum.
2- In the tested methods there are no control groups of insects. I wander what happens if due to experimental errors insects are subjected to accidental toxic substances that may cause knockdown? How reliable will be the results? The authors are invited to explain why in those tests control groups can totatlly be ignored.
The “control” group is the USDA strain that is known to be fully susceptible to phosphine. There are then five field populations all determined fully susceptible with the FAO assays, and those are predicted to be similar to the USDA strain as a known control. The remaining populations were compared to the USDA and other susceptible strains, and there were no differences in many cases. We have now properly identified the USDA strain as our experimental control as it is established as phosphine susceptible. See our addition at ll. 239-240.
Specific comments:
Line 18: …"become immobilize"… shouldn't be …"become immobilize"…?
Now changed to “immobile”, on l. 17.
Line 75: …"as 5 mg/day (>3000 ppm),"… 3000 ppm is concentration has nothing to do with the exposure time. It should be "…5 mg/m3 (>3000 ppm),…"
Terrible mistake. This value is intended as concentration, and is now corrected to “5 mg/l”.
Line 87: Schlipalius et al., 2012; is not on the list of references.
It is now added to the References.
Line 89: Schlipalius et al., 2012; is not on the list of references
It is now added to the References.
Line 216:…"… our 12 T. castaneum populations…" the 12 T. castaneum needs explanation.
We now inserted the phrase “in Table 1” to be unambiguous about what those 12 populations were, and we elaborated a bit more for clarity. See now ll. 224-227 in the Tracked Changes version..
Lines 251 and 252: The sentence "… An equal number of trials for added stimulus and no stimulus within each population were…." needs rephrasing.
We deleted that very poor sentence, and have no put “Pairs of stimulus and non-stimulus trials were performed…” as a new beginning to the sentence on ll. 256-259.
Line 273: "…ranging from 8.46-52.70…" should be : "…ranging from 8.46 to 52.70…"
Done.
Lines 278 and 279: "…in each QT assay type…" what is QT essay type? needs explanation.
Sorry, we need to be consistent. This is now clarified as “type of knockdown assay”.
Lines 324 and 326: "…KT50 …" should be "…KT50 …"
Done
Line 338: "…three strongly populations…" should be "…three strong populations…"
Changed to read “strongly resistant” to properly identify the phenotype.
Lines 445-447 "Waterford, C. J., and R. G. Winks. 1994. .." is not in text.
Now deleted from the References.

Reviewer 3 Report
In the reviewed article the authors address the problem of rapid testing and assessment of the resistance to the commonly used product stored protection agent - phosphine in populations of Tribolium castaneum. The Authors examine the effectiveness of the agent in inducing the state described by as narcosis. The Authors, using behavioral assays, try to verify the hypothesis about the legitimacy of their use to differentiate between T. castaneum lines with different degrees of resistance to fumigants. Simultaneously, the influence of the stress of tactile stimulation on the recovery of the insect from anesthesia was also verified. As the Authors admit, the test did not meet the expectations in line differentiation and the results obtained do not correlate with the currently dominant screening test. However, it seems that this direction of research is promising and there is a need to develop new tests to identify phosphine resistant populations quickly and effectively. Below is a list of remarks to the manuscript.
The brief information about the mechanism of action of the Phosphane should be included in the manuscript. Because Authors use Phosphane resistant strains of T. castaneum , description of mechanisms of resistance would be useful.
55 recommended discriminating dose of 30 ppm for 20 h is recommended
The unit of “ppm” is a concentration unit, but Authors use this unit as a dose. In my opinion, this is not a proper use of the term in this case. Mixed use of concentration or dose appears many times in the manuscript.
57 The discriminating dose is the estimated concentration needed to kill 99.9%
The concentration is notoriously called a "dose" which is incorrect. The dose is the amount of substance that the organism intakes. This remark concerns the entire manuscript.
75 compared to concentrations used for pest control, they will enter narcosis
In the manuscript Authors frequently use the term narcosis which I consider incorrect. In my opinion, the commonly used term “tonic immobility” (TI, death-feigning) is more appropriate in the given context (Nishi et al. 2010, Kiyotake et al. 2014, Nakayama and Miyatake 2009, Kim et al. 2018).
89 at two different loci (Schlipalius et al., 2012).
It is not listed in the Bibliography.
91 The RR50 is computed from dose-mortality data subjected to Probit analysis on a given population for which the LD50 in the resistant population is divided by the LD50 for a standard lab-susceptible population.
It should be an LC50 unit instead of LD50.
Fig1.
I suggest adding the letters denoting described figures (1. A, B, C) for the clarity (Generally, quality of photos should be improved)
Tab 1.
I would appreciate an explanation why the Standard Error (SE) was used instead of Standard Deviation (SD)?
Fig 3.
Why Authors conducted only the statistical comparison within each population but not between susceptible, resistant?
178 replications with multiple beetles, this helped to control for possible error that could occur when one beetle is left to be knocked down, and as soon as it becomes motionless, a different beetle within the replication regains mobility. We found that the behavior of alternating between moving and not moving created uncertainty in the observer about whether the test insect was actually completely “motionless” at that point the observation was recorded.
It seems that the source of the issue may be the experimental set-up and the measurement methods themselves. Much better standardization could be achieved by recording the behavior of insects and then performing automatic analysis. Also, the longitudinal, inclined test tube makes the distribution of insects, and the interaction between them is uneven (which is visible in Fig 2).
Materials and methods section general remarks:
What were the conditions for conducting tests and observations? In this respect, the illumination, RH and temperature will have a significant influence on the results, how were these conditions controlled?
It seems that in the case of "reaction on stimulus - single test" there is no adequate control (positive and negative), so it is difficult to conclude on the results as the authors do. Perhaps it would be worthwhile to place 1 immobilized insect with 9 dead or 9 alive individuals and compare the recovery times of this single insect? This way it would be possible to verify to what extent the mechanical stimulation itself and to what extent the presence of other insects influences the outcome.
Table 1
In my opinion, it's in the wrong place. These are the results and should be placed in the results section. Therefore, some of the materials and methods section should be transferred to the results. The description of the single/group test should also be modified for the sake of legibility. Additionally, I am not certain if you can compare the results of the one versus many insects tests, or KT50 and KT100, these are entirely different parameters. It is similar to statistically compare the values of e.g. LC50 and LC95 .
There are no GC results. Since there is information in the materials and methods that GC tests were performed to verify the actual concentrations in the test tubes, the results of these analyses should also be included.
Author Response
Dear Reviewer 3,
My responses to your comments and corrections are in the attached Word file. I have also pasted a copy of the same file below, as a security measure. Thank you so much for your helpful review.
Tom Phillips
********************
Reviewer 3
Comments and Suggestions for Authors
In the reviewed article the authors address the problem of rapid testing and assessment of the resistance to the commonly used product stored protection agent - phosphine in populations of Tribolium castaneum. The Authors examine the effectiveness of the agent in inducing the state described by as narcosis. The Authors, using behavioral assays, try to verify the hypothesis about the legitimacy of their use to differentiate between T. castaneum lines with different degrees of resistance to fumigants. Simultaneously, the influence of the stress of tactile stimulation on the recovery of the insect from anesthesia was also verified. As the Authors admit, the test did not meet the expectations in line differentiation and the results obtained do not correlate with the currently dominant screening test. However, it seems that this direction of research is promising and there is a need to develop new tests to identify phosphine resistant populations quickly and effectively. Below is a list of remarks to the manuscript.
The brief information about the mechanism of action of the Phosphane should be included in the manuscript. Because Authors use Phosphane resistant strains of T. castaneum , description of mechanisms of resistance would be useful.
We have now added a sentence to the Introductions about phosphine mode of action and the potential for evolution of resistance. Now in lines 43-46.
55 recommended discriminating dose of 30 ppm for 20 h is recommended
The unit of “ppm” is a concentration unit, but Authors use this unit as a dose. In my opinion, this is not a proper use of the term in this case. Mixed use of concentration or dose appears many times in the manuscript.
The term “dose” appears in our manuscript 7 times. We understand the reviewer’s point that “concentration” of the toxic gas is a more accurate description, but we ask that the editors allow us to keep the term “dose”. Six of the occurrences are in the phrase “discriminating dose assay”, a practice in resistance toxicology that uses the minimum amount of a toxin that kills all susceptible individuals, but that should be too low to kill resistant insects. It is a standard resistance term we would like ot retain. The seventh use was in the phrase “dose response assays”, also a common term in toxicology, but replaced it with “toxicity assays” in line 393.
57 The discriminating dose is the estimated concentration needed to kill 99.9%
The concentration is notoriously called a "dose" which is incorrect. The dose is the amount of substance that the organism intakes. This remark concerns the entire manuscript.
As discussed above, we request that we retain the term “dose” in those 6 cases where it is part of the generic name for a procedure call the “discriminating dose test”, which is a common practice for detecting insecticide resistance.
75 compared to concentrations used for pest control, they will enter narcosis
In the manuscript Authors frequently use the term narcosis which I consider incorrect. In my opinion, the commonly used term “tonic immobility” (TI, death-feigning) is more appropriate in the given context (Nishi et al. 2010, Kiyotake et al. 2014, Nakayama and Miyatake 2009, Kim et al. 2018).
We used the word “narcosis” 8 times in our paper. This term is widely used in the insect fumigation literature related to this response. The earliest works we cite are the papers that first used narcosis for a fumigant response not resulting in death. The insect therefore “passes out”, goes into a narcotic state, from which it usually always “wakes up”, or is revived, after some time in clean air after the toxic gas is vented away. A similar term is “knockdown”, used widely for a resistance response in various insects. We use “knockdown” in the manuscript’s title and for all of our results. The insect is knocked down, is immobilized, by the very high concentration of the toxic gas, but then is revived. Our consensus was that “narcosis” can be interpreted by readers in ways we do not intend, so we chose “knockdown” for describing our work. “Tonic immobility” is an interesting response in various animals. Wikipedia defines it as:
“… a natural state of paralysis that animals enter, often called animal hypnosis. Its function is not certain. It may be related to mating in certain animals like sharks. It may also be a way of avoiding or deterring predators (playing dead is called thanatosis)”
I am not aware of this term used in insecticide toxicology.
89 at two different loci (Schlipalius et al., 2012).
It is not listed in the Bibliography.
Our error. It is now in the References section.
91 The RR50 is computed from dose-mortality data subjected to Probit analysis on a given population for which the LD50 in the resistant population is divided by the LD50 for a standard lab-susceptible population.
It should be an LC50 unit instead of LD50.
Agreed. We changed it to “LC” on those lines. We appreciate the reviewer pointing this out, because it certainly is a misuse of the term “dose” for a fumigation experiment analyzed with Probit analysis.
Fig1.
I suggest adding the letters denoting described figures (1. A, B, C) for the clarity (Generally, quality of photos should be improved.
Good point. The figure is being revised with additional advice form the journal.
Tab 1.
I would appreciate an explanation why the Standard Error (SE) was used instead of Standard Deviation (SD)?
The reporting of an SE along with the mean value it describes is a very common practice in applied entomology journals. SE is calculated from the observed SD divided by the square root of the experiment’s sample size. We had observations (n) of either 10 or 20 observations in our experiments that were randomly drawn from our “lab population” of many 1000s of beetles for each strain. Wikipedia reports that “.. SE is used… to make confidence intervals of the unknown population mean.” We will never practically know the true population mean for our beetle populations. However, the idea is that, as we approach a true mean for our true population (as n gets much closer to a large number), then the SE and SD become nearly identical. Therefore, SE is an estimate of the SD for the true population. I will of course defer to the journal on this.
Fig 3.
Why Authors conducted only the statistical comparison within each population but not between susceptible, resistant?
There are two experiments reported in this figure, one for a strain of susceptible beetles and the other for a strain of resistant beetles. Each had an “untreated control”, for which the tube was not rolled (hence “no disturbance” was the control), compared to the “treatment” of having “disturbance” of the single beetle by having the tube rolled 360 degrees in each direction at the start of successive minutes of gas exposure. We selected these two strains to see if disturbance, as would be experienced in beetles tested in a group, for each of them would prolong the time until knockdown. We found disturbance in fact had that effect for both, and we see clearly that the magnitude between control and treatment was much larger for the resistant strain tested. We made a few revisions in the narrative of the Methods (ll. 298-307) to provide more clarity on this.
178 replications with multiple beetles, this helped to control for possible error that could occur when one beetle is left to be knocked down, and as soon as it becomes motionless, a different beetle within the replication regains mobility. We found that the behavior of alternating between moving and not moving created uncertainty in the observer about whether the test insect was actually completely “motionless” at that point the observation was recorded.
It seems that the source of the issue may be the experimental set-up and the measurement methods themselves. Much better standardization could be achieved by recording the behavior of insects and then performing automatic analysis. Also, the longitudinal, inclined test tube makes the distribution of insects, and the interaction between them is uneven (which is visible in Fig 2).
Many thanks to the reviewer for raising these questions. The photo in Fig. 2 shows some of the devices used in the bioassay method, but it is not correct for an accurate description of how the bioassays were actually conducted. We have revised the text and removed that photo.
Materials and methods section general remarks:
What were the conditions for conducting tests and observations? In this respect, the illumination, RH and temperature will have a significant influence on the results, how were these conditions controlled?
We have now added information
It seems that in the case of "reaction on stimulus - single test" there is no adequate control (positive and negative), so it is difficult to conclude on the results as the authors do. Perhaps it would be worthwhile to place 1 immobilized insect with 9 dead or 9 alive individuals and compare the recovery times of this single insect? This way it would be possible to verify to what extent the mechanical stimulation itself and to what extent the presence of other insects influences the outcome.
Good point. We could have done more in the way of having both positive and negative controls. The addition of 9 dead beetles is a very good idea which would get at the different biological variables at work in “knockdown” tests. Unfortunately, we are not able to repeat or conduct new experiment at this time. We have added the idea of additional conspecifics, whether causing a disturbance or not, affecting knockdown times of other beetles. See the discussion, ll. 380-389.
Table 1
In my opinion, it's in the wrong place. These are the results and should be placed in the results section. Therefore, some of the materials and methods section should be transferred to the results. The description of the single/group test should also be modified for the sake of legibility. Additionally, I am not certain if you can compare the results of the one versus many insects tests, or KT50 and KT100, these are entirely different parameters. It is similar to statistically compare the values of e.g. LC50 and LC95 .
Agreed. The paper needs proper formatting. The current arrangement was made by the Journal, The placement of figures and tables relative to Methods and Results will be optimized in the final draft.
As per our statistical comparison among the Single vs KT50 and KT100 data for each population, we confirmed that we used the proper testing via Two-Way ANOVA followed by means comparison tests. We first wanted to evaluate three different methods to analyze 12 beetle populations for resistance. We already know by the independently performed FAO tests that the first 6 are susceptible and the last 6 are resistant. So, we compared the 12 populations for each of the three quick test methods, which required a separate statistical comparison for each of the three. The null hypothesis for each method was that there would be no difference in knockdown time among the 12 populations. All three tests showed differences among populations, but none showed clear separation between susceptible and resistant populations.
Then as a second type of comparison we investigated which test could be completed in the shortest time for any given population by comparing the times needed to conclude a resistance status in each population, so 12 separate comparison going across. Interestingly, the single and KT50 had similar times in all but one population; the KT 100 was different from the other two in all but one population. The three resistance tests are very different, as you say, but there results were scored with the same variable: time. To clarify this more we added a new sentence at ll. 238-240.
There are no GC results. Since there is information in the materials and methods that GC tests were performed to verify the actual concentrations in the test tubes, the results of these analyses should also be included.
Our GC analyses were done to ensure methods were standardized for exposure to 3000 ppm in our tubes. The lengthy paragraph on GC Methods is now deleted and a shortened method for applying gas and measuring its concentration is now given on ll.212-213.

Reviewer 4 Report
This manuscript investigates “Evaluation of Knockdown Bioassay Methods to Assess Phosphine Resistance in the Red Flour Beetle,
Tribolium castaneum (Herbst) (Coleoptera: Tenebrionidae)". This information is of interest to toxicologist and entomologists working within insecticide resistance. The experimental set up of this study appears to be well-designed and the data collected carefully. However, the authors did not clarify the statistical methods to explain some results obtained in the experiments. In addition, some methods in the manuscript are incomplete. Also, introduction and methods section should be summarized. I think that this manuscript requires substantial rewriting to make its results clearer and more readily interpretable to the reader. My specific comments are listed in the "Main text". The authors provide knowledge in this manuscript and can be of great interest to the journal. Based on the comments above reported, my opinion is that this manuscript may be suitable for printing on this journal.

Author Response
Dear Reviewer 4,
My responses to your questions and comments are in the attached Word file. I have copied and pasted the same file below, as a security measure. Thank you so much for your very helpful review.
Tom Phillips
**********
Reviewer 4
This manuscript investigates “Evaluation of Knockdown Bioassay Methods to Assess Phosphine Resistance in the Red Flour Beetle,
Tribolium castaneum (Herbst) (Coleoptera: Tenebrionidae)". This information is of interest to toxicologist and entomologists working within insecticide resistance. The experimental set up of this study appears to be well-designed and the data collected carefully. However, the authors did not clarify the statistical methods to explain some results obtained in the experiments. In addition, some methods in the manuscript are incomplete. Also, introduction and methods section should be summarized. I think that this manuscript requires substantial rewriting to make its results clearer and more readily interpretable to the reader. My specific comments are listed in the "Main text". The authors provide knowledge in this manuscript and can be of great interest to the journal. Based on the comments above reported, my opinion is that this manuscript may be suitable for printing on this journal.
We appreciate these first comments by the reviewer. As per comments from other reviewers we have made revisions throughout the newest draft that address the concerns raised here on experimental methods and statistical analyses.
Response to specific questions and suggested revisions placed in the “Main Text” sent as a PDF are answered below by line number.
We have revised the various strikethrough of targeted words or phrases as needed.
L. 34 Keywords should be in alphabetic order.
Done.
L. 38 and throughout: For all references cited, check according to the journal style (place references in numerical order).
Done, as well as citations all by number, and references listed in numerical order.
ll. 68-84 Summarize this paragraph
Done; see smaller paragraph at ll. 73-91
ll. 85-104 Summarize
Done; see smaller paragraph at ll. 92-113.
ll. 128-136. Paragraph have various objectives, please, summarize or combine.
Done.
l. 161. Please, provide more information about statistical analysis in materials and methods section
The statistics were originally described in the sections explaining methods for each experiment. We now have a new section in the Methods at line 273 that clearly describes the statistics that were done.
l. 162. Knockdown definition should be summarized.
Done. The length of this paragraph on “Knockdown definition” is now greatly reduced.
ll. 400, 423, 452, 453. Abbreviate name of journal/publication.
Done
Round 2
Reviewer 3 Report
I have no more comments. All my questions and suggestions were explained by Authors.
Reviewer 4 Report
The authors have incorporated all suggestions and comments into the revised version, now the ms seems much clear. There are some minor points to be corrected:
Line 28: place “;” after management
Line 40: place “.” after [5-8]
Line 68: change “mg/l” by “mg/L”
Line 69: change “cthey” by “they”
Line 141: pull apart “for30”
Line 181: delete “.”
Line 184: add “.”
Line 185: delete “.”
Line 221: delete “.”
Line 234: delete “.”
Lines 352 and 370: T. castanaeum should be in italic